# Explanatory Models of Burnout Diagnosis Based on Personality Factors in Primary Care Nurses

**DOI:** 10.3390/ijerph19159170

**Published:** 2022-07-27

**Authors:** Luis Albendín-García, Nora Suleiman-Martos, Elena Ortega-Campos, Raimundo Aguayo-Estremera, José L. Romero-Béjar, Guillermo A. Cañadas-De la Fuente

**Affiliations:** 1Casería de Montijo Health Center, Granada-Metropolitan Health District, Andalusian Health Service, Calle Virgen de la Consolación, 12, 18015 Granada, Spain; lualbgar1979@ugr.es; 2Faculty of Health Sciences, University of Granada, Avenida de la Ilustración, 60, 18016 Granada, Spain; norasm@ugr.es (N.S.-M.); gacf@ugr.es (G.A.C.-D.l.F.); 3Department of Psychology, University of Almería, 04120 Almería, Spain; elenaortega@ual.es; 4Department of Psychobiology and Methodology in Behavioral Sciences, Complutense University of Madrid, Campus de Somosaguas, 28223 Pozuelo de Alarcón, Spain; raaguayo@ucm.es; 5Department of Statistics and Operations Research, University of Granada, Fuentenueva s/n, 18071 Granada, Spain; 6Instituto de Investigación Biosanitaria (ibs. GRANADA), 18012 Granada, Spain; 7Institute of Mathematics of the University of Granada (IMAG), Ventanilla 11, 18001 Granada, Spain; 8Brain, Mind and Behaviour Research Center (CIMCYC), University of Granada, 18071 Granada, Spain

**Keywords:** personality risk factors, logistic regression, burnout, nurses, primary care

## Abstract

Burnout in the primary care service takes place when there is a high level of interaction between nurses and patients. Explanatory models based on psychological and personality related variables provide an approximation to level changes in the three dimensions of the burnout syndrome. A categorical-response ordinal logistic regression model, based on a quantitative, crosscutting, multicentre, descriptive study with 242 primary care nurses in the Andalusian Health Service in Granada (Spain) is performed for each dimension. The three models included all the variables related to personality. The risk factor *friendliness* was significant at population level for the three dimensions, whilst *openness* was never significant. *Neuroticism* was significant in the models related to emotional exhaustion and depersonalization, whilst *responsibility* was significant for the models referred to depersonalization and low personal accomplishment dimensions. Finally, *extraversion* was also significant in the emotional exhaustion and low personal accomplishment dimensions. The analysis performed provides useful information, making more readily the diagnosis and evolution of the burnout syndrome in this collective.

## 1. Introduction

Primary care (PC) is the first level of access for the population to the Andalusian public health system and is characterized by providing comprehensive health care. PC includes preventive, curative, and rehabilitative care, as well as health promotion, health education, and environmental health surveillance. These elements are affected by the current pandemic situation, increasing the care load in health centers [1,2]. The PC setting is characterized by the presence of numerous psychosocial and occupational stressors related to patient and family care in the community framework. In addition, these stressors are also affected by personality variables. Therefore, quantifying their effects for prognosis in a worsening level of each dimension of burnout to know the status of the professionals who carry out their care activity in PC centers is justified [3,4].

Burnout syndrome is a problem with a high prevalence among those who provide services to the public, something that has drawn the attention of many researchers [5,6]. Care practice development of health professionals could be affected as a consequence of these stressors, which in turn can cause burnout, a syndrome characterized by the presence of emotional exhaustion (EE) or physical overexertion and emotional fatigue as a result of interaction with users; depersonalization (DP) or cynical treatment toward these users; and low personal accomplishment (PA) or loss of confidence and negative self-concept due to unrewarding situations.

Burnout is related to work stress, which negatively impacts the professionals involved and their workplace [7]. The term ‘burnout’ was first used in 1974 by Freudenberger and since then many authors have tried to describe the stress experienced by workers in the workplace that becomes chronic [8]. Research on burnout since then has shown that this syndrome has a particular impact among workers whose occupations require a lot of social interaction with those who use their services. Initially, this problem was addressed in the field of social services, but over time a growing number of professions have been identified in which symptoms arise that fit within the theoretical framework of burnout syndrome. In fact, health professionals are very affected by this syndrome and nursing, in particular, has been the subject of many studies because the risk of presenting burnout is very high [4].

Currently and from a psychology perspective, the most accepted definition of burnout syndrome is the one described by Maslach and Jackson [9]. This definition is based on the previously mentioned three-dimensional syndrome and it would be the result of poor adaptation to the work environment. This form of conceptualising the syndrome is based on the Maslach Burnout Inventory (MBI), which is the most commonly used means of assessing burnout [10]. However, from a biomedical point of view and according to the World Health Organization, burnout is defined as the result of a situation of chronic stress in the workplace [11], is included in the ICD-11, and is even recognized as an occupational disease in some countries such as Sweden and the Netherlands [12,13].

Within health professions, the syndrome is especially frequent among nurses, with PC nurses [14] being especially vulnerable. Numerous studies have been carried out to analyze the influence of factors such as age, parenthood, length of employment, shift work, and workload in the development of burnout [15,16]. Burnout can have serious consequences. Different negative effects of this syndrome have been described in health professionals, such as insomnia, irritability, or drug use, among others [17,18]. Adverse effects of burnout have also been identified in health institutions, such as an increase in absenteeism, work incapacity, and an increase in treatment errors, which affects the quality of care for users of the health system [19]. The main signs and symptoms of burnout among PC nurses include fatigue, difficulty concentrating, poor organization, higher number of errors, decreased quality of work, lack of energy, anxiety, and frustration. When these circumstances occur and persist for a significant period, they can lead to the appearance of burnout [3,4,14].

It should be noted that analyzing the most significant psychological variables to predict the severity of burnout is especially useful in the management of PC nursing services [20,21]. This implies carrying out correct health promotion so that nurses increase control over their own health. Therefore, this study has the purpose (1) to identify risk factors related to personality variables that allow explaining the different models of severity of burnout and (2) quantify their effect for prognosis at the different levels of each dimension of the burnout syndrome for PC nurses.

## 2. Materials and Methods

### 2.1. Design and Procedure

A cross-sectional multicenter study was made in Andalusia (Spain), with a sample of PC nurses in this region. The nurses were contacted by volunteer collaborators from the Spanish nursing union (SATSE), who helped to receive the completed questionnaires.

### 2.2. Participants

The sample comprised 242 nurses working at the Andalusian Health Service. The mean age of the subjects was 46.5 years (SD = 7.45); 55.0% of them were female. A total of 90.3% of the nurses worked a fixed morning shift and 65.1% worked on-call duties. The mean duration of their current position was 133.6 ± 101.6 months, and in the profession, it was 282.47 ± 91.1 months.

### 2.3. Variables and Instruments

An ad hoc questionnaire was used to obtain data on socio-demographic variables (age and sex), the three dimensions of burnout (EE, DP and PA), and five personality variables that were assessed using the Spanish version of the NEO Five-Factor Inventory (NEO-FFI) [22]. The personality variables were neuroticism (Nt) or the level of emotional instability; extraversion (Ex) or the level of energy and sociability; friendliness (Fr) or the level of interpersonal tendencies of approach or rejection to others; responsibility (Ry) or the level of self-control and self-determination; and openness (Op) or the level of intellectual curiosity and aesthetic sensibility. The NEO-FFI consists of 60 items. Each personality factor is assessed using 12 items, scored on a five-point Likert scale. The final score for each factor is the sum of its 12 items. The Cronbach alpha for Nt is 0.92, 0.89 for Ex, 0.86 for Fr, 0.90 for Ry, and 0.87 for Op.

The dimensions of burnout were measured with the Spanish version of the Maslach Burnout Inventory (MBI). The MBI consists of 9 items for EE, 5 items for DP, and 8 items for PA, up to a total of 22. The Cronbach alpha for EE is 0.89, 0.77 for D,, and 0.78 for PA. The diagnostic values used to establish high, medium or low levels of each dimension, EE (low: <19, medium: 19–26, high: >26), D (low: <6, medium: 6–9, high: >9), and PA (low: <34, medium: 34–39, high: >39), were those proposed by manual test [23].

### 2.4. Ethics

This study was approved by the Ethics Committee of the University of Granada (393/CEIH2017) and carried out in accordance with the ethical standards of the Declaration of Helsinki [24]. The nurses received information about the study and gave their written consent. Participation in the study was voluntary, individual, and anonymous.

### 2.5. Statistical Methods

First, a graphical bivariate and tri-variate exploratory analysis was performed to identify which pairs and triplet of variables were able to separate, in worsening levels, for each dimension of burnout syndrome. Second, a categorical-response ordinal logistic regression model was used [25,26] for each one of the dimensions of burnout, considering psychological and personality-related data as explanatory variables. These models were used to determine which variables caused transitions among levels of burnout. A model containing the effects of the factors, with no interaction between them, was considered to best fit the data. This model was fitted in a stepwise way starting from a constant model, using forward selection to determine whether a variable enters, and backward selection to determine whether it exits, in each step. The goodness-of-fit was compared using the likelihood ratio test and Pearson’s chi-squared test. The statistical significance of the parameters for the variables that enter into each model was evaluated using Wald’s test and the prognosis ratios for each level with respect to the adjacent level were obtained, depending on the possible changes in the explanatory variables considered. Statistical analyses were performed using the R Statistical Computing Software (version 4.1.1).

## 3. Results

### 3.1. Description of the Sample and Levels of the Three Dimensions of Burnout

The descriptive analysis of the personality variables considered is shown in Table 1.

The scores for the three dimensions of burnout syndrome were categorized as low, medium, or high, according to the indications of the MBI. The 29.8% of the participants in this sample had high EE, whilst 25.2% presented a medium score for this dimension. On the other hand, 38.8% had high scores for DP and 28.5% had a medium score in this level. Finally, 29.3% presented high score for PA and 31.4% obtained a medium score. Table 2 shows the prevalence of the levels for each dimension of burnout.

### 3.2. Exploratory Analysis

The potential utility as classifiers, in a worsening level for each dimension of burnout, is explored for each pair of personality variables jointly. Figure 1, Figure 2 and Figure 3 represent high levels for each burnout dimension in blue, medium levels in green, and low levels in red. It is reflected that different pairs of variables seem to be useful to separate high levels for each dimension of burnout with respect to medium and low levels. However, none of these pairs are able to separate low and medium levels. This is because the blue points seem to separate from those of green and red colours, but not between these two other colours. For instance, if Figure 1 is considered as a matrix with five rows and five columns, the graphical output in position (row = 1, column = 2), that faces neuroticism vs. friendliness, shows how blue points are separated from those of green and red colors and therefore, this pair of variables seems to be useful to identify high levels of emotional exhaustion dimension of burnout syndrome.

A similar analysis is performed for triplets of variables. In this case, it should be noted that the variables neuroticism, friendliness, and extraversion, jointly, seem to be useful to identify high levels of emotional exhaustion (see Figure 4). On the other hand, variables neuroticism, friendliness, and responsibility, jointly, provide a good classification in high levels of depersonalization dimension of burnout (see Figure 5). Finally, in Figure 6 it is reflected that variables friendliness, responsibility, and extraversion are adequate to identify high levels of low personal accomplishment dimension. This is because the blue points in these graphical outputs seem to separate from those of green and red colours. Indeed, this is consistent with the results obtained in the section below since these are the variables, which included in the models for each dimension of burnout, are significant at population level.

### 3.3. Explanatory Model for Each Dimension of Burnout

The estimated model for each dimension of burnout includes the five explanatory variables related to personality according to the following form:
(1)
$${\hat{L}}_{s}\left(\mathrm{Nt},\mathrm{Fr},\mathrm{Ry},\mathrm{Ex},\mathrm{Op}\right)={\hat{B}}_{0}+{\hat{B}}_{\mathrm{Nm}}\mathrm{Nt}+{\hat{B}}_{\mathrm{Fr}}\mathrm{Fr}+{\hat{B}}_{\mathrm{Ry}}\mathrm{Ry}+{\hat{B}}_{\mathrm{Ex}}\mathrm{Ex}+{\hat{B}}_{\mathrm{Op}}\mathrm{Op};s=1.2$$


The parameters estimated for each explanatory variable in the ordinal logistic regression model, for each dimension of burnout, are shown in Table 3, Table 4 and Table 5 below.

The log-likelihood test for these models were X^2^ (5, *n* = 242) = 114.57, *p* < 0.05 for emotional exhaustion (EE); X^2^ (5, *n* = 242) = 81.854, *p* ≤ 0.05 for depersonalization (DP); and X^2^ (5, *n* = 242) = 102.625, *p* ≤ 0.05 for low personal accomplishment (PA). Therefore, when these variables were included in the model, the fit improved significantly compared to a model than only takes the constant into account. The Pearson chi-square goodness-of-fit test for these models were X^2^ (473, *n* = 258) = 911.72, *p* < 0.05 for EE; X^2^ (473, *n* = 258) = 872.347, *p* < 0.05 for DP; and X^2^ (473, *n* = 258) = 102.625, *p* < 0.05 for PA. These results concluded, therefore, that the models produced a good fit at population level with all dimensions.

In light of the results of the Wald test (see Table 3, Table 4 and Table 5), the variable friendliness (Fr) is significant at population level for the three dimensions (*p* < 0.001), whilst openness (Op) is never significant (*p* > 0.05). Neuroticism (Nm) is significant in the models related to emotional exhaustion (*p* < 0.001) and depersonalization (*p* < 0.001), whilst responsibility (Ry) is significant for the models referred to depersonalization (*p* = 0.042) and low personal accomplishment (*p* < 0.001) dimensions. Finally, extraversion (Ex) is significant in the models estimated to emotional exhaustion (*p* < 0.014) and low personal accomplishment dimensions (*p* = 0.01).

Odds ratios (see Table 3, Table 4 and Table 5) for the significant variables can be interpreted as measures of strength related to increasing or decreasing severity in each dimension. All of the considered personality-related variables are discrete variables; therefore, it is not expected for relevant changes to occur with just one unit of increase in these variables (the odds ratios are close to one), but with more units of increase. In fact, if Fr increased by six units, the odds ratio referred to as moving to a low level of EE (to a low level of DP and PA) would be 1.45 (1.63 and 1.47, respectively) times greater than if Fr did not increase. The opposite effect took place with Nt in EE and DP. Thus, if Nt decreased by six units, the odds ratio of passing to a low level of EE (DP) would be 2.09 (1.4). Finally, an increase of six units in the value of Ry produced an odds ratio of moving to a low level of 1.83 times greater for PA and 1.33 times greater for DP, whilst an increase of six units in the value of Ex involved odds ratios of passing to low levels of 1.39 greater for both PA and EE. It is important to highlight that none of the confidence intervals of the odds ratios for changes of six units in the significant variables contained a value of 1.

## 4. Discussion

The objectives of this study were to identify risk factors related to personality variables and quantify their effects for prognosis at the different levels of each dimension of burnout syndrome for PC nurses. With regard to the first objective and the personality-related variables, three models that provide a first approximation of level changes in each of the three dimensions of burnout syndrome were obtained. These models included all of the variables related to personality as explanatory variables. Friendliness is a protective factor included in the three models, and consequently it is involved in level changes for the three dimensions of burnout syndrome. Ry is included in the models related to depersonalization and low personal accomplishment; therefore, it is a relevant variable involved in level changes for these two dimensions. Ex is involved in level changes associated with the dimensions related to EE and PA, whilst Nt is involved in the models related to EE and DP being relevant in their level changes. These models are able to predict the probability of an individual being at a burnout level, according to changes in the explanatory variables. With regard to the second objective, the results showed that high values of Fr were associated with situations of lower burnout severity in the three dimensions of this syndrome. In the same way, Ry is a protective factor involving decreasing burnout severity in the DP and PA dimensions, and Ex is also a protection factor in the same way in the EE and PA dimensions. In contrast, higher values of Nt were associated with increasing burnout severity in the EE and DP dimensions.

Nurses characterized by Fr suffer less CE, less DP, and greater RP. This protective factor is due to the fact that they have higher energy, empathize with patients and help them to be more comfortable, and feel more fulfilled with their work [21,27,28]. Neurotic people have a high tendency to instability and psychological distress [29]. In the relationship that exists between Nt and EE, several authors state that emotional instability may be the consequence of the stressful care pressure to which nurses are subjected [21,30,31]. If this condition becomes chronic due to poor coping [32], it can result in other health problems [33] and even abandonment of work [34].

Ry in professional nursing times is inherent in the fact of feeling that they must contribute to a common good, especially in troubled times such as the COVID-19 pandemic [35]. The excess of Ry can decrease PA due to problems of motivation and job satisfaction and due to a lack of training [36]. To this excess of Ry should be added the lack of human and material resources as well as long working hours, which ends up affecting productivity and makes it difficult to meet their work objectives [20,37,38,39]. For this reason, it is very important to encourage teamwork to prevent this type of issue [40].

The Ex is a personal protective factor against burnout, since people who have this type of positive traits tend to use effective coping strategies, decreasing EE and increasing PA [41]. This also contributes to a positive work environment, resulting in an increase in care quality [42]. For this reason, it is convenient for institutions to support nursing staff so that they acquire coping skills and promote Ex [21]. These are based on mature coping skills based on humor, suppression, anticipation, and sublimation [29]. Otherwise, job dissatisfaction, poor performance, and development of burnout would appear [43].

### 4.1. Study Limitations

The limitations of this study are related to the methodological design. This cross-sectional study does not allow us to establish causal relationships in our results. It would be advisable to analyze new results in a prospective longitudinal design. The evolution of burnout in nursing professionals of primary health care could be analyzed. Another limitation to consider is that the sampling was for convenience in order to focus on these professionals in particular.

### 4.2. Clinical Implications

PC nurses are essential in the early care of patients. Precisely for this reason, there is a great demand for care that overloads professionals with work. If the lack of human and material resources is added to this, job dissatisfaction ends up taking its toll and affects the physical and psychological health of health personnel [44].

This study confirms that the appearance of burnout is determined by the work performed and by the personality of the nurse. In addition, changes were identified in the personality variables that can make nurses predisposed to suffer from burnout and which personality variables have more weight. The fact of knowing which personality variables are the most important to predict changes in the severity of burnout syndrome, as has been achieved with other variables, would be very useful in the management of nursing units in primary health care [45].

For this reason, knowing the current situation allows establishing health promotion and prevention measures that prevent the appearance of the syndrome [46]. Therefore, intervention measures that strengthen coping techniques are recommended [47,48].

## 5. Conclusions

Future nursing professionals who will work and who already work in primary health care are exposed to burnout syndrome. The personality variables that relate to professionals in this care area include high levels of Nt and Ry and low levels of Ex and Fr. It is advisable to take this profile into account to better meet the needs of nursing staff.

PC managers must be able to identify risk profiles and provide material and human resources that allow quality care work to be carried out. They should also promote coping strategies in PC nurses.

## Figures and Tables

**Figure 1 ijerph-19-09170-f001:**
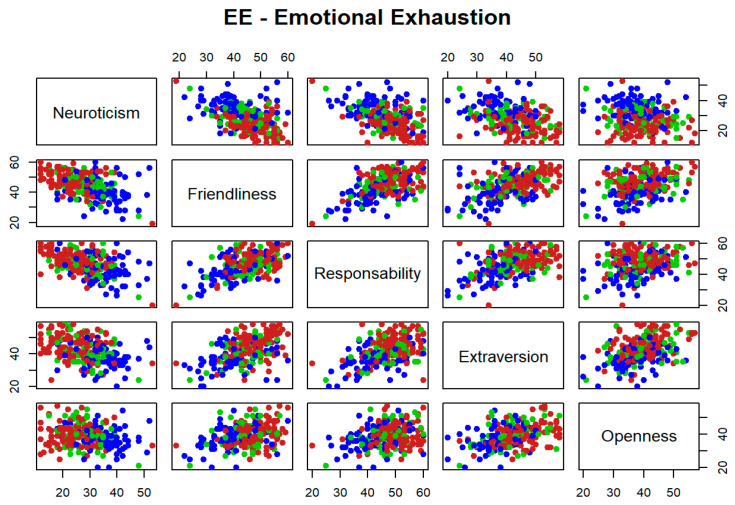
Bivariate exploratory analysis for Emotional Exhaustion.

**Figure 2 ijerph-19-09170-f002:**
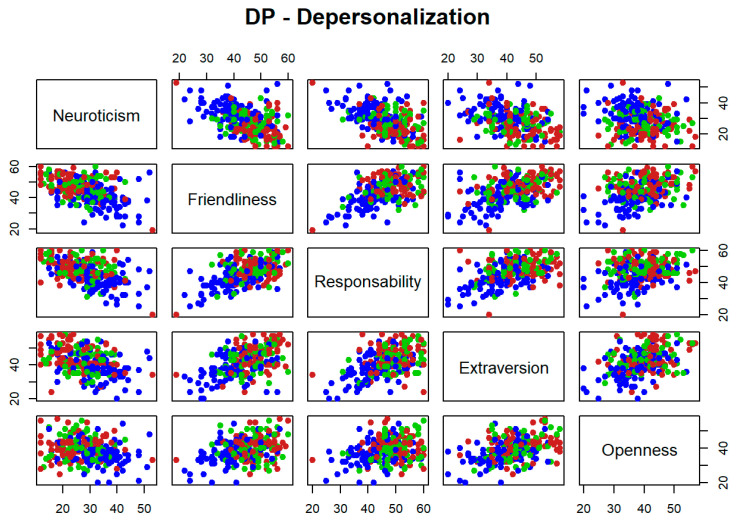
Bivariate exploratory analysis for Depersonalization.

**Figure 3 ijerph-19-09170-f003:**
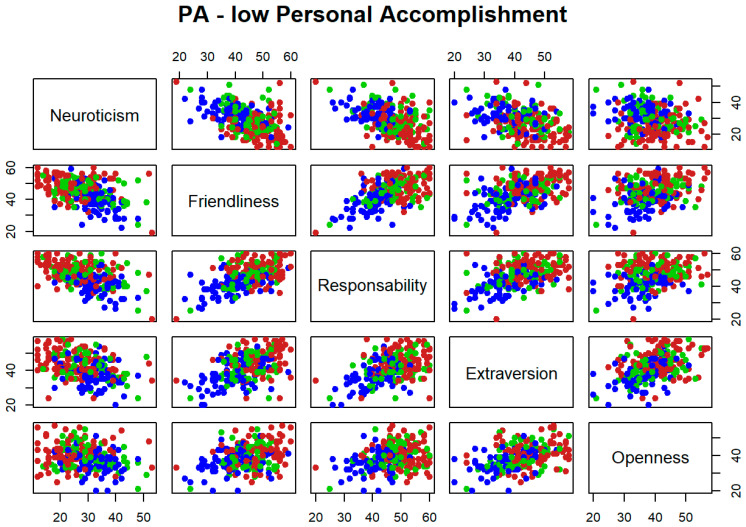
Bivariate exploratory analysis for low Personal Accomplishment.

**Figure 4 ijerph-19-09170-f004:**
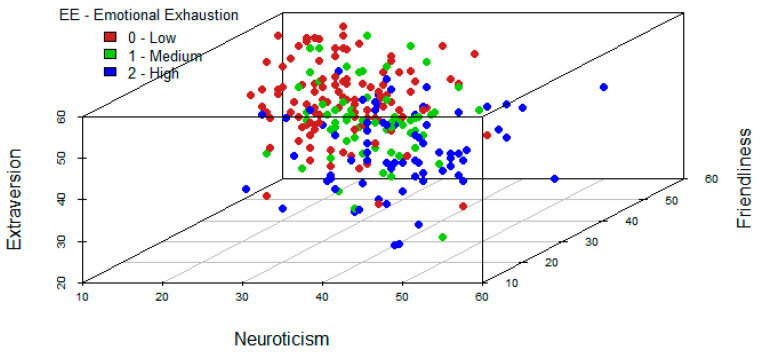
Tri-variate exploratory analysis for Emotional Exhaustion.

**Figure 5 ijerph-19-09170-f005:**
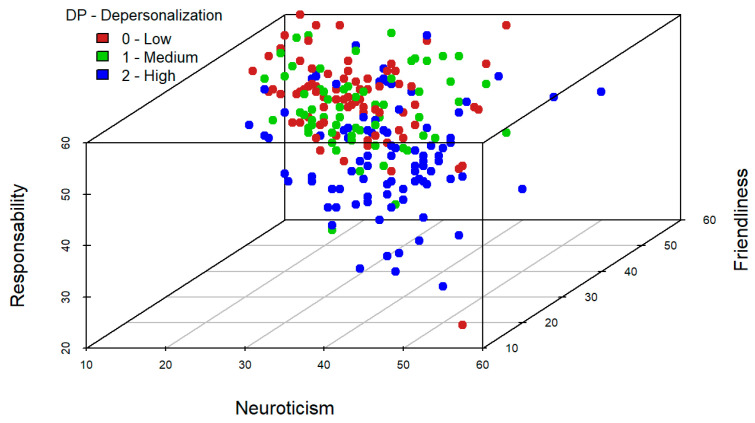
Tri-variate exploratory analysis for Depersonalization.

**Figure 6 ijerph-19-09170-f006:**
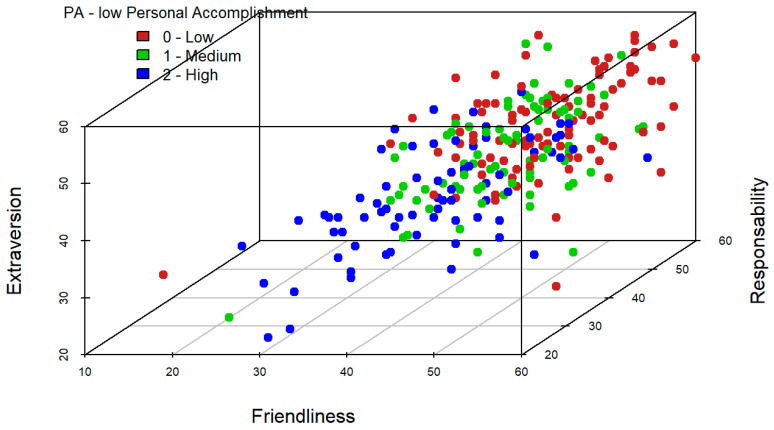
Tri-variate exploratory analysis for low Personal Accomplishment.

**Table 1 ijerph-19-09170-t001:** Descriptive analysis of the personality variables.

Variable (*n* = 242)	M	SD
Nt	28.26	8.20
Fr	44.72	7.65
Ry	46.60	7.22
Ex	42.01	7.88
Op	39.08	6.72

Note: Nt = neuroticism, Fr = friendliness, Ry = responsibility, Ex = extraversion, Op = openness, M = mean, SD = standard deviation.

**Table 2 ijerph-19-09170-t002:** Levels of burnout for each dimension.

Levels	EE			DP			PA		
	Low	Medium	High	Low	Medium	High	Low	Medium	High
%	45.0	25.2	29.8	32.7	28.5	38.8	39.3	31.4	29.3
M (SD)	18.44 (12.24)			7.69 (6.29)			36.40 (8.92)		

Note: EE = emotional exhaustion, DP = depersonalisation, PA = low personal accomplishment, M = mean, SD = standard deviation.

**Table 3 ijerph-19-09170-t003:** Logit model for Emotional Exhaustion (EE).

Predictor	B	SD	Wald	*p*	Odds	CI for 95% Odds	
						Lower	Upper
EE=1	−1.951	1.698	1.321	0.250	0.142	0.005	3.961
EE=2	−0.349	1.692	0.043	0.837	0.705	0.025	19.423
Nt	0.123	0.022	30.123	<0.001	1.131	1.082	1.182
Fr	−0.063	0.024	6.902	<0.001	0.939	0.896	0.984
Ry	−0.031	0.025	1.506	0.220	0.969	0.923	1.018
Ex	−0.055	0.022	5.996	0.014	0.946	0.905	0.989
Op	0.037	0.023	2.541	0.111	1.038	0.991	1.086

Note: EE = emotional exhaustion, Nt = neuroticism, Fr = friendliness, Ry = responsibility, Ex = extraversion, Op = openness, B = estimated parameter, SD = standard deviation, Wald = Wald statistic, *p* = *p*-value, Odds = odds ratio, CI = confidence interval, Lower = lower limit of the CI, Upper = upper limit of the CI.

**Table 4 ijerph-19-09170-t004:** Logit model for Depersonalization (DP).

Predictor	B	SD	Wald	*p*	Odds	CI for 95% Odds	
						Lower	Upper
D=1	−6.898	1.715	16.185	<0.001	0.001	<0.001	0.03
D=2	−5.350	1.691	10.007	<0.001	0.004	<0.001	0.131
Nt	0.057	0.020	8.082	<0.001	1.058	1.018	1.100
Fr	−0.082	0.023	12.379	<0.001	0.921	0.880	0.964
Ry	−0.048	0.024	4.135	0.042	0.953	0.909	0.998
Ex	−0.014	0.021	0.481	0.488	0.986	0.946	1.026
Op	−0.026	0.022	1.389	0.239	0.974	0.933	1.017

Note: EE = emotional exhaustion, Nm = neuroticism, Fr = friendliness, Ry = responsibility, Ex = extraversion, Op = openness, B = estimated parameter, SD = standard deviation, Wald = Wald statistic, *p* = *p*-value, Odds = odds ratio, CI = confidence interval, Lower = lower limit of the CI, Upper = upper limit of the CI.

**Table 5 ijerph-19-09170-t005:** Logit model for low Personal Accomplishment (PA).

Predictor	B	SD	Wald	*p*	Odds	CI for 95% Odds	
						Lower	Upper
PA=1	−10.215	1.809	31.871	<0.001	<0.001	<0.001	0.001
PA=2	−8.350	1.766	22.359	<0.001	<0.001	<0.001	0.007
Nt	0.022	0.020	1.181	0.277	1.021	0.982	1.062
Fr	−0.065	0.023	7.790	<0.001	0.937	0.895	0.980
Ry	−0.100	0.025	16.057	<0.001	0.904	0.861	0.950
Ex	−0.055	0.021	6.567	0.010	0.946	0.908	0.987
Op	−0.008	0.022	0.129	0.719	0.992	0.949	1.036

Note: EE = emotional exhaustion, Nt = neuroticism, Fr = friendliness, Ry = responsibility, Ex = extraversion, Op = openness, B = estimated parameter, SD = standard deviation, Wald = Wald statistic, *p* = *p*-value, Odds = odds ratio, CI = confidence interval, Lower = lower limit of the CI, Upper = upper limit of the CI.

## Data Availability

Not applicable.

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
