# Peer review of "Explanatory Models of Burnout Diagnosis Based on Personality Factors in Primary Care Nurses"

_ijerph, 2022, doi:10.3390/ijerph19159170_

Round 1

Reviewer 1 Report

The study is timely and the authors have taken a unique approach to factors associated with burnout. I have a few comments to improve the paper. 

- Do you have any additional demographics such as years of experience, years on the particular unit, etc? These would be great to compare burnout levels across different groups. 

- What were the psychometric properties of the instrument for this sample? There needs to be at least a reliability coefficient for each scale. Is there any evidence of validity in previous studies? 

- Why is the DP scale scored differently than the other scales? It seems that a lower score on this scale means more of the attribute. Is that true?

- Figures 1-3 and 4-6 all need a more detailed explanation on how to interpret these figures. For readers who are not statistically adept, there needs to be an example of how you arrived at your results you reported (e.g., different pairs of variables seem to be useful to separate high levels for each dimension of burnout with respect to medium and low levels. p4, line 169-171). It would be helpful to at least explain the first figure and how you read the results and then do the same for figures 4-6. 

- The logistic regression is well done and clear. 

Author Response

Please, see file attached.

Reviewer 2 Report

The research topic and method were intriguing for reading the paper.

1. Participants

Please indicate the basis and appropriateness of the selection of the number of subjects.

2. Variables and Instruments

The reliability of the tool used for variable measurement is not given.

3. Results

Unfortunately, the characteristics of the subjects' socio-demographic variables are not shown. It is insufficient to grasp the socio-demographic characteristics of the subjects only by the ratio of average age and gender.

4. Table 1. Descriptive analysis of the personality variables needs to be supplemented. It is difficult to understand the personality variables because only the simple mean and standard deviation are presented.

5. Table 2. Each element of burnout syndrome is categorized as low, medium, and high, and the reference value needs to be presented.

6. You have concluded that personality variations are related to professional nurses. It would be nice to suggest the possible interventions for the management to apply.

Author Response

Please, see attached file.
